# EFFICIENT LIFELONG LEARNING WITH A-GEM

**Arslan Chaudhry[1], Marc'Aurelio Ranzato[2], Marcus Rohrbach[2], Mohamed Elhoseiny[2]**
[1]University of Oxford, [2]Facebook AI Research
`arslan.chaudhry@eng.ox.ac.uk`, `{ranzato,mrf,elhoseiny}@fb.com`

## ABSTRACT

In lifelong learning, the learner is presented with a sequence of tasks, incrementally building a data-driven prior which may be leveraged to speed up learning of a new task. In this work, we investigate the *efficiency* of current lifelong approaches, in terms of sample complexity, computational and memory cost. Towards this end, we first introduce a new and a more realistic evaluation protocol, whereby learners observe each example only once and hyper-parameter selection is done on a small and disjoint set of tasks, which is not used for the actual learning experience and evaluation. Second, we introduce a new metric measuring how quickly a learner acquires a new skill. Third, we propose an improved version of GEM (Lopez-Paz & Ranzato, 2017), dubbed Averaged GEM (A-GEM), which enjoys the same or even better performance as GEM, while being almost as computationally and memory efficient as EWC (Kirkpatrick et al., 2016) and other regularization-based methods. Finally, we show that all algorithms including A-GEM can learn even more quickly if they are provided with task descriptors specifying the classification tasks under consideration. Our experiments on several standard lifelong learning benchmarks demonstrate that A-GEM has the best trade-off between accuracy and efficiency.[1]

## 1 INTRODUCTION

Intelligent systems, whether they are natural or artificial, must be able to quickly adapt to changes in the environment and to quickly learn new skills by leveraging past experiences. While current learning algorithms can achieve excellent performance on a variety of tasks, they strongly rely on copious amounts of supervision in the form of labeled data.

The *lifelong learning* (LLL) setting attempts at addressing this shortcoming, bringing machine learning closer to a more realistic human learning by acquiring new skills quickly with a small amount of training data, given the experience accumulated in the past. In this setting, the learner is presented with a stream of tasks whose relatedness is not known a priori. The learner has then the potential to learn more quickly a new task, if it can remember how to combine and re-use knowledge acquired while learning related tasks of the past. Of course, for this learning setting to be useful, the model needs to be constrained in terms of amount of compute and memory required. Usually this means that the learner should not be allowed to merely store all examples seen in the past (in which case this reduces the lifelong learning problem to a multitask problem) nor should the learner be engaged in computations that would not be feasible in real-time, as the goal is to quickly learn from a stream of data.

Unfortunately, the established training and evaluation protocol as well as current algorithms for lifelong learning do not satisfy all the above desiderata, namely learning from a stream of data using limited number of samples, limited memory and limited compute. In the most popular training paradigm, the learner does several passes over the data (Kirkpatrick et al., 2016; Aljundi et al., 2018; Rusu et al., 2016; Schwarz et al., 2018), while ideally the model should need only a handful of samples and these should be provided one-by-one in a single pass (Lopez-Paz & Ranzato, 2017). Moreover, when the learner has several hyper-parameters to tune, the current practice is to go over the sequence of tasks several times, each time with a different hyper-parameter value, again ignoring the requirement of learning from a stream of data and, strictly speaking, violating the assumption of

---

the LLL scenario. While some algorithms may work well in a single-pass setting, they unfortunately require a lot of computation (Lopez-Paz & Ranzato, 2017) or their memory scales with the number of tasks (Rusu et al., 2016), which greatly impedes their actual deployment in practical applications.

In this work, we propose an evaluation methodology and an algorithm that better match our desiderata, namely learning efficiently – in terms of training samples, time and memory – from a stream of tasks. First, we propose a new learning paradigm, whereby the learner performs cross validation on a set of tasks which is disjoint from the set of tasks actually used for evaluation (Sec. 2). In this setting, the learner will have to learn and will be tested on an entirely new sequence of tasks and it will perform just a single pass over this data stream. Second, we build upon GEM (Lopez-Paz & Ranzato, 2017), an algorithm which leverages a small episodic memory to perform well in a single pass setting, and propose a small change to the loss function which makes GEM orders of magnitude faster at training time while maintaining similar performance; we dub this variant of GEM, A-GEM (Sec. 4). Third, we explore the use of compositional task descriptors in order to improve the few-shot learning performance within LLL showing that with this additional information the learner can pick up new skills more quickly (Sec. 5). Fourth, we introduce a new metric to measure the speed of learning, which is useful to quantify the ability of a learning algorithm to learn a new task (Sec. 3). And finally, using our new learning paradigm and metric, we demonstrate A-GEM on a variety of benchmarks and against several representative baselines (Sec. 6). Our experiments show that A-GEM has a better trade-off between average accuracy and computational/memory cost. Moreover, all algorithms improve their ability to quickly learn a new task when provided with compositional task descriptors, and they do so better and better as they progress through the learning experience.

## 2 LEARNING PROTOCOL

Currently, most works on lifelong learning (Kirkpatrick et al., 2016; Rusu et al., 2016; Shin et al., 2017; Nguyen et al., 2018) adopt a learning protocol which is directly borrowed from supervised learning. There are $T$ tasks, and each task consists of a training, validation and test sets. During training the learner does as many passes over the data of each task as desired. Moreover, hyper-parameters are tuned on the validation sets by sweeping over the whole sequence of tasks as many times as required by the cross-validation grid search. Finally, metrics of interest are reported on the test set of each task using the model selected by the previous cross-validation procedure.

Since the current protocol violates our stricter definition of LLL for which the learner can only make a single pass over the data, as we want to emphasize the importance of learning quickly from data, we now introduce a new learning protocol.

We consider two streams of tasks, described by the following ordered sequences of datasets $\mathcal{D}^{CV} = \{\mathcal{D}_1, \cdots, \mathcal{D}_{T^{CV}}\}$ and $\mathcal{D}^{EV} = \{\mathcal{D}_{T^{CV}+1}, \cdots, \mathcal{D}_T\}$, where $\mathcal{D}_k = \{(\mathbf{x}_i^k, t_i^k, y_i^k)_{i=1}^{n_k}\}$ is the dataset of the $k$-th task, $T^{CV} < T$ (in all our experiments $T^{CV} = 3$ while $T = 20$), and we assume that all datasets are drawn from the same distribution over tasks. To avoid cluttering of the notation, we let the context specify whether $\mathcal{D}_k$ refers to the training or test set of the $k$-th dataset.

$\mathcal{D}^{CV}$ is the stream of datasets which will be used during cross-validation; $\mathcal{D}^{CV}$ allows the learner to replay all samples multiple times for the purposes of model hyper-parameter selection. Instead, $\mathcal{D}^{EV}$ is the actual dataset used for final training and evaluation on the test set; the learner will observe training examples from $\mathcal{D}^{EV}$ once and only once, and all metrics will be reported on the test sets of $\mathcal{D}^{EV}$. Since the regularization-based approaches for lifelong learning (Kirkpatrick et al., 2016; Zenke et al., 2017) are rather sensitive to the choice of the regularization hyper-parameter, we introduced the set $\mathcal{D}^{CV}$, as it seems reasonable in practical applications to have similar tasks that can be used for tuning the system. However, the actual training and testing are then performed on $\mathcal{D}^{EV}$ using a single pass over the data. See Algorithm 1 for a summary of the training and evaluation protocol.

Each example in any of these dataset consists of a triplet defined by an input ($\mathbf{x}^k \in \mathcal{X}$), task descriptor ($t^k \in \mathcal{T}$, see Sec. 5 for examples) and a target vector ($y^k \in \mathbf{y}^k$), where $\mathbf{y}^k$ is the set of labels specific to task $k$ and $\mathbf{y}^k \subset \mathcal{Y}$. While observing the data, the goal is to learn a predictor $f_\theta : \mathcal{X} \times \mathcal{T} \to \mathcal{Y}$, parameterized by $\theta \in \mathbb{R}^P$ (a neural network in our case), that can map any test pair $(\mathbf{x}, t)$ to a target $y$.

---

**Algorithm 1** Learning and Evaluation Protocols

---

1: **for** $h$ **in** hyper-parameter list **do**                ▷ Cross-validation loop, executing multiple passes over $\mathcal{D}^{CV}$
2:     **for** $k = 1$ **to** $T^{CV}$ **do**                            ▷ Learn over data stream $\mathcal{D}^{CV}$ using $h$
3:         **for** $i = 1$ **to** $n_k$ **do**                        ▷ Single pass over $\mathcal{D}_k$
4:             Update $f_\theta$ using $(\mathbf{x}_i^k, t_i^k, y_i^k)$ and hyper-parameter $h$
5:             Update metrics on test set of $\mathcal{D}^{CV}$
6:         **end for**
7:     **end for**
8: **end for**
9: Select best hyper-parameter setting, $h^*$, based on average accuracy of test set of $\mathcal{D}^{CV}$, see Eq. 1.
10: Reset $f_\theta$.
11: Reset all metrics.
12: **for** $k = T^{CV} + 1$ **to** $T$ **do**                            ▷ Actual learning over datastream $\mathcal{D}^{EV}$
13:     **for** $i = 1$ **to** $n_k$ **do**                        ▷ Single pass over $\mathcal{D}_k$
14:         Update $f_\theta$ using $(\mathbf{x}_i^k, t_i^k, y_i^k)$ and hyper-parameter $h^*$
15:         Update metrics on test set of $\mathcal{D}^{EV}$
16:     **end for**
17: **end for**
18: Report metrics on test set of $\mathcal{D}^{EV}$.

---

## 3   METRICS

Below we describe the metrics used to evaluate the LLL methods studied in this work. In addition to Average Accuracy ($A$) and Forgetting Measure ($F$) (Chaudhry et al., 2018), we define a new measure, the Learning Curve Area (LCA), that captures how quickly a model learns.

The training dataset of each task, $\mathcal{D}_k$, consists of a total $B_k$ mini-batches. After each presentation of a mini-batch of task $k$, we evaluate the performance of the learner on all the tasks using the corresponding test sets. Let $a_{k,i,j} \in [0, 1]$ be the accuracy evaluated on the test set of task $j$, after the model has been trained with the $i$-th mini-batch of task $k$. Assuming the first learning task in the continuum is indexed by 1 (it will be $T^{CV} + 1$ for $\mathcal{D}^{EV}$) and the last one by $T$ (it will be $T^{CV}$ for $\mathcal{D}^{CV}$), we define the following metrics:

**Average Accuracy** ($A \in [0, 1]$)   Average accuracy after the model has been trained continually with all the mini-batches up till task $k$ is defined as:

$$A_k = \frac{1}{k} \sum_{j=1}^{k} a_{k, B_k, j} \tag{1}$$

In particular, $A_T$ is the average accuracy on all the tasks after the last task has been learned; this is the most commonly used metric used in LLL.

**Forgetting Measure** ($F \in [-1, 1]$)   (Chaudhry et al., 2018) Average forgetting after the model has been trained continually with all the mini-batches up till task $k$ is defined as:

$$F_k = \frac{1}{k-1} \sum_{j=1}^{k-1} f_j^k \tag{2}$$

where $f_j^k$ is the forgetting on task '$j$' after the model is trained with all the mini-batches up till task $k$ and computed as:

$$f_j^k = \max_{l \in \{1, \cdots, k-1\}} a_{l, B_l, j} - a_{k, B_k, j} \tag{3}$$

Measuring forgetting after all tasks have been learned is important for a two-fold reason. It quantifies the accuracy drop on past tasks, and it gives an indirect notion of how quickly a model may learn a new task, since a forgetful model will have little knowledge left to transfer, particularly so if the new task relates more closely to one of the very first tasks encountered during the learning experience.

**Learning Curve Area (LCA $\in [0,1]$)**    Let us first define an average $b$-shot performance (where $b$ is the mini-batch number) after the model has been trained for all the $T$ tasks as:

$$Z_b = \frac{1}{T} \sum_{k=1}^{T} a_{k,b,k} \tag{4}$$

LCA at $\beta$ is the area of the convergence curve $Z_b$ as a function of $b \in [0, \beta]$:

$$\text{LCA}_\beta = \frac{1}{\beta + 1} \int_0^\beta Z_b db = \frac{1}{\beta + 1} \sum_{b=0}^{\beta} Z_b \tag{5}$$

LCA has an intuitive interpretation. $\text{LCA}_0$ is the average 0-shot performance, the same as forward transfer in Lopez-Paz & Ranzato (2017). $\text{LCA}_\beta$ is the area under the $Z_b$ curve, which is high if the 0-shot performance is good and if the learner learns quickly. In particular, there could be two models with the same $Z_\beta$ or $A_T$, but very different $\text{LCA}_\beta$ because one learns much faster than the other while they both eventually obtain the same final accuracy. This metric aims at discriminating between these two cases, and it makes sense for relatively small values of $\beta$ since we are interested in models that learn from few examples.

## 4    AVERAGED GRADIENT EPISODIC MEMORY (A-GEM)

So far we discussed a better training and evaluation protocol for LLL and a new metric to measure the speed of learning. Next, we review GEM (Lopez-Paz & Ranzato, 2017), which is an algorithm that has been shown to work well in the single epoch setting. Unfortunately, GEM is very intensive in terms of computational and memory cost, which motivates our efficient variant, dubbed A-GEM. In Sec. 5, we will describe how compositional task descriptors can be leveraged to further speed up learning in the few shot regime.

GEM avoids catastrophic forgetting by storing an episodic memory $\mathcal{M}_k$ for each task $k$. While minimizing the loss on the current task $t$, GEM treats the losses on the episodic memories of tasks $k < t$, given by $\ell(f_\theta, \mathcal{M}_k) = \frac{1}{|\mathcal{M}_k|} \sum_{(\mathbf{x}_i, k, y_i) \in \mathcal{M}_k} \ell(f_\theta(\mathbf{x}_i, k), y_i)$, as inequality constraints, avoiding their increase but allowing their decrease. This effectively permits GEM to do positive backward transfer which other LLL methods do not support. Formally, at task $t$, GEM solves for the following objective:

$$\text{minimize}_\theta \quad \ell(f_\theta, \mathcal{D}_t) \quad \text{s.t.} \quad \ell(f_\theta, \mathcal{M}_k) \leq \ell(f_\theta^{t-1}, \mathcal{M}_k) \qquad \forall k < t \tag{6}$$

Where $f_\theta^{t-1}$ is the network trained till task $t - 1$. To inspect the increase in loss, GEM computes the angle between the loss gradient vectors of previous tasks $g_k$, and the proposed gradient update on the current task $g$. Whenever the angle is greater than 90° with any of the $g_k$'s, it projects the proposed gradient to the closest in L2 norm gradient $\tilde{g}$ that keeps the angle within the bounds. Formally, the optimization problem GEM solves is given by:

$$\text{minimize}_{\tilde{g}} \quad \frac{1}{2} \|g - \tilde{g}\|_2^2 \quad \text{s.t.} \quad \langle \tilde{g}, g_k \rangle \geq 0 \qquad \forall k < t \tag{7}$$

Eq.7 is a quadratic program (QP) in $P$-variables (the number of parameters in the network), which for neural networks could be in millions. In order to solve this efficiently, GEM works in the dual space which results in a much smaller QP with only $t - 1$ variables:

$$\text{minimize}_v \quad \frac{1}{2} v^\top G G^\top v + g^\top G^\top v \quad \text{s.t.} \quad v \geq 0 \tag{8}$$

where $G = -(g_1, \cdots, g_{t-1}) \in \mathbb{R}^{(t-1) \times P}$ is computed at each gradient step of training. Once the solution $v^*$ to Eq. 8 is found, the projected gradient update can be computed as $\tilde{g} = G^\top v^* + g$.

While GEM has proven very effective in a single epoch setting (Lopez-Paz & Ranzato, 2017), the performance gains come at a big computational burden at training time. At each training step, GEM computes the matrix $G$ using all samples from the episodic memory, and it also needs to solve the QP of Eq. 8. Unfortunately, this inner loop optimization becomes prohibitive when the size of $\mathcal{M}$ and the number of tasks is large, see Tab. 7 in Appendix for an empirical analysis. To alleviate

the computational burden of GEM, next we propose a much more efficient version of GEM, called Averaged GEM (A-GEM).

Whereas GEM ensures that at every training step the loss of each *individual* previous tasks, approximated by the samples in episodic memory, does not increase, A-GEM tries to ensure that at every training step the *average* episodic memory loss over the previous tasks does not increase. Formally, while learning task $t$, the objective of A-GEM is:

$$\text{minimize}_\theta \quad \ell(f_\theta, \mathcal{D}_t) \quad \text{s.t.} \quad \ell(f_\theta, \mathcal{M}) \leq \ell(f_\theta^{t-1}, \mathcal{M}) \qquad \text{where } \mathcal{M} = \cup_{k<t}\mathcal{M}_k \qquad (9)$$

The corresponding optimization problem reduces to:

$$\text{minimize}_{\tilde{g}} \quad \frac{1}{2}||g - \tilde{g}||_2^2 \quad \text{s.t.} \quad \tilde{g}^\top g_{ref} \geq 0 \qquad (10)$$

where $g_{ref}$ is a gradient computed using a batch randomly sampled from the episodic memory, $(\mathbf{x}_{ref}, y_{ref}) \sim \mathcal{M}$, of all the past tasks. In other words, A-GEM replaces the $t-1$ constraints of GEM with a single constraint, where $g_{ref}$ is the average of the gradients from the previous tasks computed from a random subset of the episodic memory.

The constrained optimization problem of Eq. 10 can now be solved very quickly; when the gradient $g$ violates the constraint, it is projected via:

$$\tilde{g} = g - \frac{g^\top g_{ref}}{g_{ref}^\top g_{ref}} g_{ref} \qquad (11)$$

The formal proof of the update rule of A-GEM (Eq. 11) is given in Appendix C. This makes A-GEM not only memory efficient, as it does not need to store the matrix $G$, but also orders of magnitude faster than GEM because 1) it is not required to compute the matrix $G$ but just the gradient of a random subset of memory examples, 2) it does not need to solve any QP but just an inner product, and 3) it will incur in less violations particularly when the number of tasks is large (see Tab. 7 and Fig. 6 in Appendix for empirical evidence). All together these factors make A-GEM faster while not hampering its good performance in the single pass setting.

Intuitively, the difference between GEM and A-GEM loss functions is that GEM has better guarantess in terms of worst-case forgetting of each individual task since (at least on the memory examples) it prohibits an increase of any task-specific loss, while A-GEM has better guaratees in terms of average accuracy since GEM may prevent a gradient step because of a task constraint violation although the overall average loss may actually decrease, see Appendix Sec. D.1 and D.2 for further analysis and empirical evidence. The pseudo-code of A-GEM is given in Appendix Alg. 2.

## 5 JOINT EMBEDDING MODEL USING COMPOSITIONAL TASK DESCRIPTORS

In this section, we discuss how we can improve forward transfer for all the LLL methods including A-GEM. In order to speed up learning of a new task, we consider the use of compositional task descriptors where components are shared across tasks and thus allow transfer. Examples of compositional task descriptors are, for instance, a natural language description of the task under consideration or a matrix specifying the attribute values of the objects to be recognized in the task. In our experiments, we use the latter since it is provided with popular benchmark datasets (Wah et al., 2011; Lampert et al., 2009). For instance, if the model has already learned and remembers about two independent properties (e.g., color of feathers and shape of beak), it can quickly recognize a new class provided a descriptor specifying the values of its attributes (yellow feathers and red beak), although this is an entirely unseen combination.

Borrowing ideas from literature in few-shot learning (Lampert et al., 2014; Zhang et al., 2018; Elhoseiny et al., 2017; Xian et al., 2018), we learn a joint embedding space between image features and the attribute embeddings. Formally, let $\mathbf{x}^k \in \mathcal{X}$ be the input (e.g., an image), $t^k$ be the task descriptor in the form of a matrix of size $C_k \times A$, where $C_k$ is the number of classes in the $k$-th task and $A$ is the total number of attributes for each class in the dataset. The joint embedding model consists of a feature extraction module, $\phi_\theta : \mathbf{x}^k \to \phi_\theta(\mathbf{x}^k)$, where $\phi_\theta(\mathbf{x}^k) \in \mathbb{R}^D$, and a task embedding module, $\psi_\omega : t^k \to \psi_\omega(t^k)$, where $\psi_\omega(t^k) \in \mathbb{R}^{C_k \times D}$. In this work, $\phi_\theta(.)$ is implemented as a standard multi-layer feed-forward network (see Sec. 6 for the exact parameterization), whereas

$\psi_\omega(.)$ is implemented as a parameter matrix of dimensions $A \times D$. This matrix can be interpreted as an attribute look-up table as each attribute is associated with a $D$ dimensional vector, from which a class embedding vector is constructed via a linear combination of the attributes present in the class; the task descriptor embedding is then the concatenation of the embedding vectors of the classes present in the task (see Appendix Fig. 9 for the pictorial description of the joint embedding model). During training, the parameters $\theta$ and $\omega$ are learned by minimizing the cross-entropy loss:

$$\ell_k(\theta, \omega) = \frac{1}{N} \sum_{i=1}^{N} -\log(p(y_i^k | \mathbf{x}_i^k, t^k; \theta, \omega)) \tag{12}$$

where $(\mathbf{x}_i^k, t^k, y_i^k)$ is the $i$-th example of task $k$. If $y_i^k = c$, then the distribution $p(.)$ is given by:

$$p(c | \mathbf{x}_i^k, t^k; \theta, \omega) = \frac{\exp([\phi_\theta(\mathbf{x}_i^k) \psi_\omega(t^k)^\top]_c)}{\sum_j \exp([\phi_\theta(\mathbf{x}_i^k) \psi_\omega(t^k)^\top]_j)} \tag{13}$$

where $[a]_i$ denotes the $i$-th element of the vector $a$. Note that the architecture and loss functions are general, and apply not only to A-GEM but also to any other LLL model (e.g., regularization based approaches). See Sec. 6 for the actual choice of parameterization of these functions.

# 6 EXPERIMENTS

We consider four dataset streams, see Tab.1 in Appendix Sec. A for a summary of the statistics. **Permuted MNIST** (Kirkpatrick et al., 2016) is a variant of MNIST (LeCun, 1998) dataset of handwritten digits where each task has a certain random permutation of the input pixels which is applied to all the images of that task. **Split CIFAR** (Zenke et al., 2017) consists of splitting the original CIFAR-100 dataset (Krizhevsky & Hinton, 2009) into 20 disjoint subsets, where each subset is constructed by randomly sampling 5 classes *without* replacement from a total of 100 classes. Similarly to Split CIFAR, **Split CUB** is an incremental version of the fine-grained image classification dataset CUB (Wah et al., 2011) of 200 bird categories split into 20 disjoint subsets of classes. **Split AWA**, on the other hand, is the incremental version of the AWA dataset (Lampert et al., 2009) of 50 animal categories, where each task is constructed by sampling 5 classes *with* replacement from the total 50 classes, constructing 20 tasks. In this setting, classes may overlap among multiple tasks, but within each task they compete against different set of classes. Note that to make sure each training example is only seen once, the training data of a each class is split into disjoint sets depending on the frequency of its occurrence in different tasks. For Split AWA, the classifier weights of each class are randomly initialized within each head without any transfer from the previous occurrence of the class in past tasks. Finally, while on Permuted MNIST and Split CIFAR we provide integer task descriptors, on Split CUB and Split AWA we stack together the attributes of the classes (specifying for instance the type of beak, the color of feathers, etc.) belonging to the current task to form a descriptor.

In terms of architectures, we use a fully-connected network with two hidden layers of 256 ReLU units each for Permuted MNIST, a reduced ResNet18 for Split CIFAR like in Lopez-Paz & Ranzato (2017), and a standard ResNet18 (He et al., 2016) for Split CUB and Split AWA. For a given dataset stream, all models use the same architecture, and all models are optimized via stochastic gradient descent with mini-batch size equal to 10. We refer to the joint-embedding model version of these models by appending the suffix '-JE' to the method name.

As described in Sec. 2 and outlined in Alg. 1, in order to cross validate we use the first 3 tasks, and then report metrics on the remaining 17 tasks after doing a single training pass over each task in sequence.

Lastly, we compared A-GEM against several baselines and state-of-the-art LLL approaches which we describe next. VAN is a single supervised learning model, trained continually without any regularization, with the parameters of a new task initialized from the parameters of the previous task. ICARL (Rebuffi et al., 2017) is a class-incremental learner that uses nearest-exemplar-based classifier and avoids catastrophic forgetting by regularizing over the feature representation of previous tasks using a knowledge distillation loss. EWC (Kirkpatrick et al., 2016), PI (Zenke et al., 2017), RWALK (Chaudhry et al., 2018) and MAS (Aljundi et al., 2018) are regularization-based approaches aiming at avoiding catastrophic forgetting by limiting learning of parameters critical to the performance of past tasks. **Progressive Networks** (PROG-NN) (Rusu et al., 2016) is a modular approach

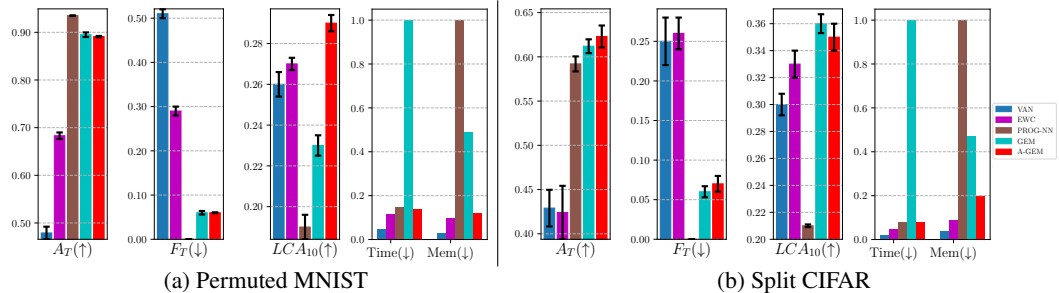

Figure 1: *Performance of LLL models across different measures on Permuted MNIST and Split CIFAR. For Accuracy ($A_T$) and Learning Curve Measure ($LCA_{10}$) the higher the number (indicated by ↑) the better is the model. For Forgetting ($F_T$), Time and Memory the lower the number (indicated by ↓) the better is the model. For Time and Memory, the method with the highest complexity is taken as a reference (value of 1) and the other methods are reported relative to that method. $A_T$, $F_T$ and $LCA_{10}$ values and confidence intervals are computed over 5 runs.* A-GEM *provides the best trade-off across different measures and dimensions. Other baselines are given in Tab. 4 and 7 in the Appendix, which are used to generate the plots.*

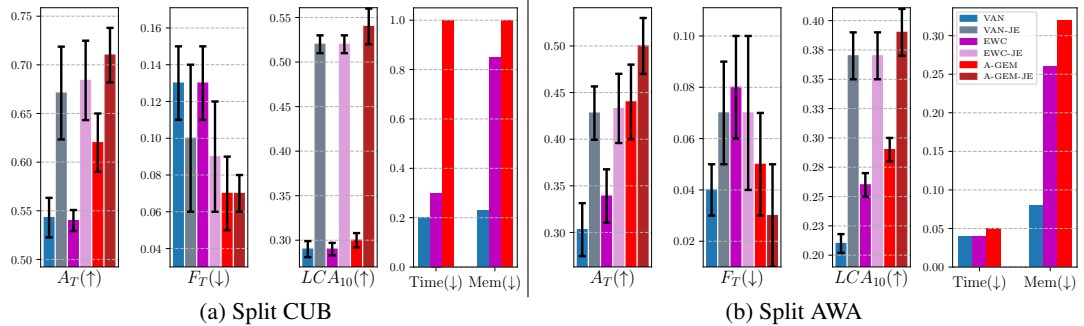

Figure 2: *Performance of LLL models across different measures on Split CUB and Split AWA. On both the datasets,* PROG-NN *runs out of memory. The memory and time complexities of joint embedding models are the same as those of the corresponding standard models and are hence omitted. $A_T$, $F_T$ and $LCA_{10}$ values and confidence intervals are computed over 10 runs. Other baselines are given in Tab. 5, 6 and 7 in the Appendix, which are used to generate the plots.*

whereby a new "column" with lateral connections to previous hidden layers is added once a new task arrives. **GEM** (Lopez-Paz & Ranzato, 2017) described in Sec. 4 is another natural baseline of comparison since A-GEM builds upon it. The amount of episodic memory per task used in ICARL, GEM and A-GEM is set to 250, 65, 50, and 100, and the batch size for the computation of $g_{ref}$ (when the episodic memory is sufficiently filled) in A-GEM is set to 256, 1300, 128 and 128 for MNIST, CIFAR, CUB and AWA, respectively. While populating episodic memory, the samples are chosen uniformly at random for each task. Whereas the network weights are randomly initialized for MNIST, CIFAR and AWA, on the other hand, for CUB, due to the small dataset size, a pre-trained ImageNet model is used. Finally, we consider a multi-task baseline, **MULTI-TASK**, trained on a single pass over shuffled data from all tasks, and thus violating the LLL assumption. It can be seen as an upper bound performance for average accuracy.

## 6.1 RESULTS

Fig. 1 and 2 show the overall results on all the datasets we considered (for brevity we show only representative methods, see detailed results in Appendix Tab. 4, 5, 6 and 7). First, we observe that A-GEM achieves the best average accuracy on all datasets, except Permuted MNIST, where PROG-NN works better. The reason is because on this dataset each task has a large number of training

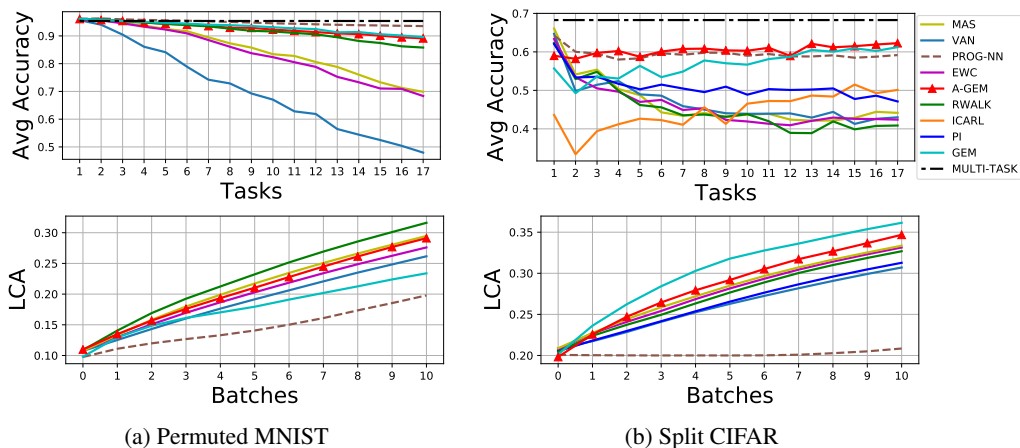

Figure 3: **Top Row:** *Evolution of average accuracy* ($A_k$) *as new tasks are learned.* **Bottom Row:** *Evolution of* LCA *during the first ten mini-batches.*

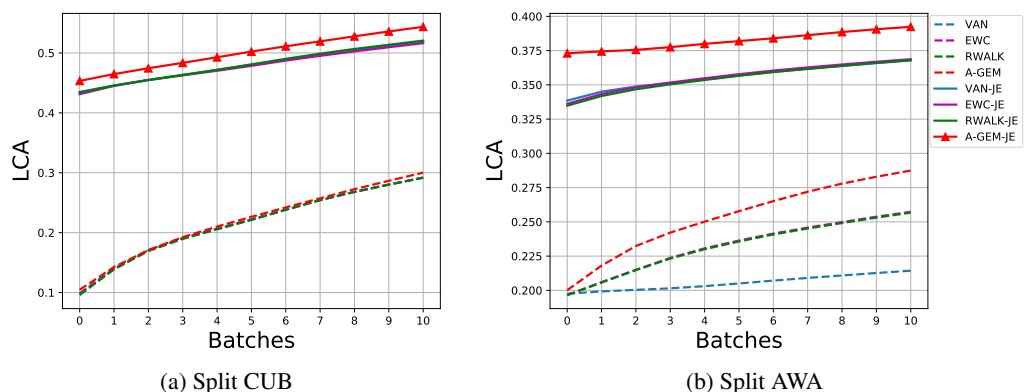

Figure 4: *Evolution of* LCA *during the first ten mini-batches.*

examples, which enables PROG-NN to learn its task specific parameters and to leverage its lateral connections. However, notice how PROG-NN has the worst memory cost by the end of training - as its number of parameters grows super-linearly with the number of tasks. In particular, in large scale setups (Split CUB and AWA), PROG-NN runs out of memory during training due to its large size. Also, PROG-NN does not learn well on datasets where tasks have fewer training examples. Second, A-GEM and GEM perform comparably in terms of average accuracy, but A-GEM has much lower time (about 100 times faster) and memory cost (about 10 times lower), comparable to regularization-based approaches like EWC. Third, EWC and similar methods perform only slightly better than VAN on this single pass LLL setting. The analysis in Appendix Sec. F demonstrates that EWC requires several epochs and over-parameterized architectures in order to work well. Fourth, PROG-NN has no forgetting by construction and A-GEM and GEM have the lowest forgetting among methods that use a fixed capacity architecture. Next, all methods perform similarly in terms of LCA, with PROG-NN being the worst because of its ever growing number of parameters and A-GEM slightly better than all the other approaches. And finally, the use of task descriptors improves average accuracy across the board as shown in Fig.2, with A-GEM a bit better than all the other methods we tried. All joint-embedding models using task descriptors have better LCA performance, although this is the same across all methods including A-GEM. Overall, we conclude that A-GEM offers the best trade-off between average accuracy performance and efficiency in terms of sample, memory and computational cost.

Fig. 3 shows a more fine-grained analysis and comparison with more methods on Permuted MNIST and Split CIFAR. The average accuracy plots show how A-GEM and GEM greatly outperform other approaches, with the exception of PROG-NN on MNIST as discussed above. On different datasets,

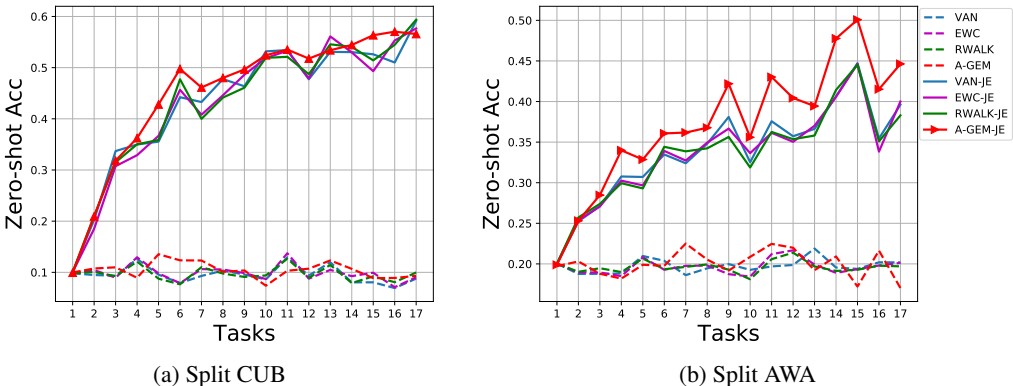

(a) Split CUB                                          (b) Split AWA

Figure 5: *Evolution of zero-shot performance as the learner sees new tasks on Split CUB and Split AWA datasets.*

different methods are best in terms of LCA, although A-GEM is always top-performing. Fig. 4 shows in more detail the gain brought by task descriptors which greatly speed up learning in the few-shot regime. On these datasets, A-GEM performs the best or on par to the best.

Finally, in Fig. 5, we report the 0-shot performance of LLL methods on Split CUB and Split AWA datasets over time, showing a clear advantage of using compositional task descriptors with joint embedding models, which is more significant for A-GEM. Interestingly, the zero-shot learning performance of joint embedding models improves over time, indicating that these models get better at forward transfer or, in other words, become more *efficient* over time.

## 7    RELATED WORK

Continual (Ring, 1997) or Lifelong Learning (LLL) (Thrun, 1998) have been the subject of extensive study over the past two decades. One approach to LLL uses modular compositional models (Fernando et al., 2017; Aljundi et al., 2017; Rosenbaum et al., 2018; Chang et al., 2018; Xu & Zhu, 2018; Ferran Alet, 2018), which limit interference among tasks by using different subset of modules for each task. Unfortunately, these methods require searching over the space of architectures which is not sample efficient with current methods. Another approach is to regularize parameters important to solve past tasks (Kirkpatrick et al., 2016; Zenke et al., 2017; Chaudhry et al., 2018), which has been proven effective for over-parameterized models in the multiple epoch setting (see Appendix Sec. F), while we focus on learning from few examples using memory efficient models. Methods based on episodic memory (Rebuffi et al., 2017; Lopez-Paz & Ranzato, 2017) require a little bit more memory at training time but can work much better in the single pass setting we considered (Lopez-Paz & Ranzato, 2017).

The use of task descriptors for LLL has already been advocated by Isele et al. (2016) but using a sparse coding framework which is not obviously applicable to deep nets in a computationally efficient way, and also by Lopez-Paz & Ranzato (2017) although they did not explore the use of compositional descriptors. More generally, tasks descriptors have been used in Reinforcement Learning with similar motivations by several others (Sutton et al., 2011; Schaul et al., 2015; Baroni et al., 2017), and it is also a key ingredient in all the zero/few-shot learning algorithms (Lampert et al., 2014; Xian et al., 2018; Elhoseiny et al., 2017; Wah et al., 2011; Lampert et al., 2009).

## 8    CONCLUSION

We studied the problem of efficient Lifelong Learning (LLL) in the case where the learner can only do a single pass over the input data stream. We found that our approach, A-GEM, has the best trade-off between average accuracy by the end of the learning experience and computational/memory cost. Compared to the original GEM algorithm, A-GEM is about 100 times faster and has 10 times less memory requirements; compared to regularization based approaches, it achieves significantly

higher average accuracy. We also demonstrated that by using compositional task descriptors all methods can improve their few-shot performance, with A-GEM often being the best.

Our detailed experiments reported in Appendix E also show that there is still a substantial performance gap between LLL methods, including A-GEM, trained in a sequential learning setting and the same network trained in a non-sequential multi-task setting, despite seeing the same data samples. Moreover, while task descriptors do help in the few-shot learning regime, the LCA performance gap between different methods is very small; suggesting a poor ability of current methods to transfer knowledge even when forgetting has been eliminated. Addressing these two fundamental issues will be the focus of our future research.

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

APPENDIX

In Sec. A we report the summary of datasets used for the experiments. Sec. B details our A-GEM algorithm and Sec. C provides the proof of update rule of A-GEM discussed in Sec. 4 of the main paper. In Sec. D, we analyze the differences between A-GEM and GEM, and describe another variation of GEM, dubbed Stochastic GEM (S-GEM). The detailed results of the experiments which were used to generate Fig 1 and 2 in the main paper are given in Sec. E. In Sec. F, we provide empirical evidence to the conjecture that regularization-based approaches like EWC require over-parameterized architectures and multiple passes over data in order to perform well as discussed in the Sec. 6.1 of the main paper. In Sec. G, we provide the grid used for the cross-validation of different hyperparameters and report the optimal values for different models. Finally, in Sec. H, we pictorially describe the joint embedding model discussed in Sec. 5.

## A  DATASET STATISTICS

Table 1: *Dataset statistics*.

|  | Perm. MNIST | Split CIFAR | Split CUB | Split AWA |
|---|---|---|---|---|
| num. of tasks | 20 | 20 | 20 | 20 |
| input size | 1×28×28 | 3×32×32 | 3×224×224 | 3×224×224 |
| num. of classes per task | 10 | 5 | 10 | 5 |
| num. of training images per task | 60000 | 2500 | 300 | - |
| num. of test images per task | 10000 | 500 | 290 | 560 |

## B  A-GEM ALGORITHM

**Algorithm 2** Training and evaluation of A-GEM on sequential data $\mathcal{D} = \{\mathcal{D}_1, \cdots, \mathcal{D}_T\}$

1: **procedure** TRAIN($f_\theta, \mathcal{D}^{train}, \mathcal{D}^{test}$)
2:   $\mathcal{M} \leftarrow \{\}$
3:   $A \leftarrow 0 \in \mathbb{R}^{T \times T}$
4:   **for** $t = \{1, \cdots, T\}$ **do**
5:     **for** $(\mathbf{x}, y) \in \mathcal{D}_t^{train}$ **do**
6:       $(\mathbf{x}_{ref}, y_{ref}) \sim \mathcal{M}$
7:       $g_{ref} \leftarrow \nabla_\theta \ell(f_\theta(\mathbf{x}_{ref}, t), y_{ref})$
8:       $g \leftarrow \nabla_\theta \ell(f_\theta(\mathbf{x}, t), y)$
9:       **if** $g^\top g_{ref} \geq 0$ **then**
10:         $\tilde{g} \leftarrow g$
11:       **else**
12:         $\tilde{g} \leftarrow g - \frac{g^\top g_{ref}}{g_{ref}^\top g_{ref}} g_{ref}$
13:       **end if**
14:       $\theta \leftarrow \theta - \alpha \tilde{g}$
15:     **end for**
16:     $\mathcal{M} \leftarrow$ UPDATEEPSMEM($\mathcal{M}, \mathcal{D}_t^{train}, T$)
17:     $A_{t,:} \leftarrow$ EVAL($f_\theta, \mathcal{D}^{test}$)
18:   **end for**
19:   **return** $f_\theta, A$
20: **end procedure**

1: **procedure** EVAL($f_\theta, \mathcal{D}^{test}$)
2:   $a \leftarrow 0 \in \mathbb{R}^T$
3:   **for** $t = \{1, \cdots, T\}$ **do**
4:     $a_t \leftarrow 0$
5:     **for** $(\mathbf{x}, y) \in \mathcal{D}_t^{test}$ **do**
6:       $a_t \leftarrow a_t +$ ACCURACY($f_\theta(\mathbf{x}, t), y$)
7:     **end for**
8:     $a_t \leftarrow \frac{a_t}{len(\mathcal{D}_t^{test})}$
9:   **end for**
10:   **return** $a$
11: **end procedure**

1: **procedure** UPDATEEPSMEM($\mathcal{M}, \mathcal{D}_t, T$)
2:   $s \leftarrow \frac{|\mathcal{M}|}{T}$
3:   **for** $i = \{1, \cdots, s\}$ **do**
4:     $(\mathbf{x}, y) \sim \mathcal{D}_t$
5:     $\mathcal{M} \leftarrow (\mathbf{x}, y)$
6:   **end for**
7:   **return** $\mathcal{M}$
8: **end procedure**

## C  A-GEM UPDATE RULE

Here we provide the proof of the update rule of A-GEM (Eq. 11), $\tilde{g} = g - \frac{g^\top g_{ref}}{g_{ref}^\top g_{ref}} g_{ref}$, stated in Sec. 4 of the main paper.

*Proof.* The optimization objective of A-GEM as described in the Eq. 10 of the main paper, is:

$$\text{minimize}_{\tilde{g}} \quad \frac{1}{2}||g - \tilde{g}||_2^2$$
$$\text{s.t.} \quad \tilde{g}^\top g_{ref} \geq 0 \tag{14}$$

Replacing $\tilde{g}$ with $z$ and rewriting Eq. 14 yields:

$$\text{minimize}_z \quad \frac{1}{2}z^\top z - g^\top z$$
$$\text{s.t.} \quad -z^\top g_{ref} \leq 0 \tag{15}$$

Note that we discard the term $g^\top g$ from the objective and change the sign of the inequality constraint. The Lagrangian of the constrained optimization problem defined above can be written as:

$$\mathcal{L}(z, \alpha) = \frac{1}{2}z^\top z - g^\top z - \alpha z^\top g_{ref} \tag{16}$$

Now, we pose the dual of Eq. 16 as:

$$\theta_\mathcal{D}(\alpha) = \min_z \mathcal{L}(z, \alpha) \tag{17}$$

Lets find the value $z^*$ that minimizes the $\mathcal{L}(z, \alpha)$ by setting the derivatives of $\mathcal{L}(z, \alpha)$ w.r.t. to $z$ to zero:

$$\nabla_z \mathcal{L}(z, \alpha) = 0$$
$$z^* = g + \alpha g_{ref} \tag{18}$$

The simplified dual after putting the value of $z^*$ in Eq. 17 can be written as:

$$\theta_\mathcal{D}(\alpha) = \frac{1}{2}(g^\top g + 2\alpha g^\top g_{ref} + \alpha^2 g_{ref}^\top g_{ref}) - g^\top g - 2\alpha g^\top g_{ref} - \alpha^2 g_{ref}^\top g_{ref}$$
$$= -\frac{1}{2}g^\top g - \alpha g^\top g_{ref} - \frac{1}{2}\alpha^2 g_{ref}^\top g_{ref}$$

The solution $\alpha^* = \max_{\alpha; \alpha > 0} \theta_\mathcal{D}(\alpha)$ to the dual is given by:

$$\nabla_\alpha \theta_\mathcal{D}(\alpha) = 0$$
$$\alpha^* = -\frac{g^\top g_{ref}}{g_{ref}^\top g_{ref}}$$

By putting $\alpha^*$ in Eq. 18, we recover the A-GEM update rule:

$$z^* = g - \frac{g^\top g_{ref}}{g_{ref}^\top g_{ref}} g_{ref} = \tilde{g}$$

$\square$

## D   ANALYSIS OF GEM AND A-GEM

In this section, we empirically analyze the differences between A-GEM and GEM, and report experiments with another computationally efficient but worse performing version of GEM.

### D.1   FREQUENCY OF CONSTRAINT VIOLATIONS

Fig. 6 shows the frequency of constraint violations (see Eq. 8 and 10) on Permuted MNIST and Split CIFAR datasets. Note that, the number of gradient updates (training steps) per task on MNIST and CIFAR are 5500 and 250, respectively. As the number of tasks increase, GEM violates the optimization constraints at almost each training step, whereas A-GEM plateaus to a much lower value. Therefore, the computational efficiency of A-GEM not only stems from the fact that it avoids solving a QP at each training step (which is much more expensive than a simple inner product) but also from the fewer number of constraint violations. From the figure, we can also infer that as the number of tasks grows the gap between GEM and A-GEM would grow further. Thus, the computational and memory overhead of GEM over A-GEM, see also Tab. 7, gets worse as the number of tasks increases.

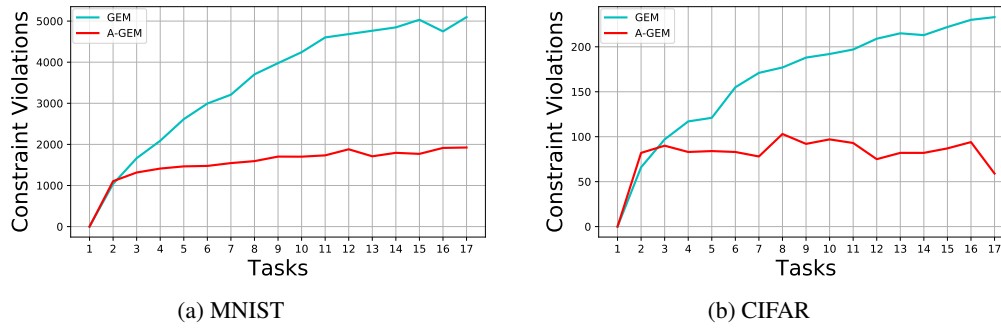

(a) MNIST

(b) CIFAR

Figure 6: *Number of constraint violations in* **GEM** *and* **A-GEM** *on* ***Permuted MNIST*** *and* ***Split CIFAR*** *as new tasks are learned.*

## D.2 AVERAGE ACCURACY AND WORST-CASE FORGETTING

In Tab. 2, we empirically demonstrate the different properties induced by the objective functions of GEM and A-GEM. GEM enjoys lower worst-case task forgetting while A-GEM enjoys better overall average accuracy. This is particularly true on the training examples stored in memory, as on the test set the result is confounded by the generalization error.

Table 2: *Comparison of average accuracy ($A_T$) and worst-case forgetting ($F_{wst}$) on the* ***Episodic Memory*** *($\mathcal{M}$) and* ***Test Set*** *($\mathcal{D}^{EV}$).*

| Methods | MNIST | | | | CIFAR | | | |
|---|---|---|---|---|---|---|---|---|
| | $\mathcal{M}$ | | $\mathcal{D}^{EV}$ | | $\mathcal{M}$ | | $\mathcal{D}^{EV}$ | |
| | $A_T$ | $F_{wst}$ | $A_T$ | $F_{wst}$ | $A_T$ | $F_{wst}$ | $A_T$ | $F_{wst}$ |
| GEM | 99.5 | 0 | 89.5 | 0.10 | 97.1 | 0.05 | 61.2 | 0.14 |
| A-GEM | 99.3 | 0.008 | 89.1 | 0.13 | 72.1 | 0.15 | 62.3 | 0.15 |

## D.3 STOCHASTIC GEM (S-GEM)

In this section we report experiments with another variant of GEM, dubbed Stochastic GEM (S-GEM). The main idea in S-GEM is to randomly sample one constraint, at each training step, from the possible $t-1$ constraints of GEM. If that constraint is violated, the gradient is projected only taking into account that constraint. Formally, the optimization objective of S-GEM is given by:

$$\text{minimize}_{\tilde{g}} \quad \frac{1}{2}||g - \tilde{g}||_2^2$$
$$\text{s.t.} \quad \langle \tilde{g}, g_k \rangle \geq 0 \quad \text{where } k \sim \{1, \cdots, t-1\} \tag{19}$$

In other words, at each training step, S-GEM avoids the increase in loss of one of the previous tasks sampled randomly. In Tab. 3 we report the comparison of GEM, S-GEM and A-GEM on Permuted MNIST and Split CIFAR.

Although, S-GEM is closer in spirit to GEM, as it requires randomly sampling one of the GEM constraints to satisfy, compared to A-GEM, which defines the constraint as the average gradient of the previous tasks, it perform slightly worse than GEM, as can be seen from Tab. 3.

Table 3: *Comparison of different variations of* GEM *on MNIST Permutations and Split CIFAR.*

| Methods | Permuted MNIST | | Split CIFAR | |
|---|---|---|---|---|
| | $A_T(\%)$ | $F_T$ | $A_T(\%)$ | $F_T$ |
| GEM | 89.5 | 0.06 | 61.2 | 0.06 |
| S-GEM | 88.2 | 0.08 | 56.2 | 0.12 |
| A-GEM | 89.1 | 0.06 | 62.3 | 0.07 |

# E   RESULT TABLES

In Tab. 4, 5, 6 and 7 we report the detailed results which were used to generate Fig.1 and 2.

Table 4: *Comparison with different baselines on Permuted MNIST and Split CIFAR. The value of $\infty$ is assigned to a metric when the model fails to train with the cross-validated values of hyper-parameters found on the subset of the tasks as discussed in Sec. 2 of the main paper. The numbers are averaged across 5 runs using a different seed each time. The results from this table are used to generate Fig 1 in Sec. 6.1 of the main paper.*

| Methods | Permuted MNIST | | | Split CIFAR | | |
|---|---|---|---|---|---|---|
| | $A_T(\%)$ | $F_T$ | $LCA_{10}$ | $A_T(\%)$ | $F_T$ | $LCA_{10}$ |
| VAN | 47.9 (± 1.32) | 0.51 (± 0.01) | 0.26 (± 0.006) | 42.9 (± 2.07) | 0.25 (± 0.03) | 0.30 (± 0.008) |
| ICARL | - | - | - | 50.1 | 0.11 | - |
| EWC | 68.3 (± 0.69) | 0.29 (± 0.01) | 0.27 (± 0.003) | 42.4 (± 3.02 ) | 0.26 (± 0.02) | 0.33 (± 0.01) |
| PI | $\infty$ | $\infty$ | $\infty$ | 47.1 (± 4.41) | 0.17 (± 0.04) | 0.31 (± 0.008) |
| MAS | 69.6 (± 0.93) | 0.27 (± 0.01) | 0.29 (± 0.003) | 44.2 (± 2.39) | 0.25 (± 0.02) | 0.33 (± 0.009) |
| RWALK | 85.7 (± 0.56) | 0.08 (± 0.01) | **0.31** (± 0.005) | 40.9 (± 3.97) | 0.29 (± 0.04) | 0.32 (± 0.005) |
| PROG-NN | **93.5** (± 0.07) | **0** | 0.19 (± 0.006) | 59.2 (± 0.85) | **0** | 0.21 (± 0.001) |
| GEM | 89.5 (± 0.48) | 0.06 (± 0.004) | 0.23 (± 0.005) | 61.2 (± 0.78) | 0.06 (± 0.007) | **0.36** (± 0.007) |
| **A-GEM (Ours)** | 89.1 (± 0.14) | 0.06 (± 0.001) | 0.29 (± 0.004) | **62.3** (± 1.24) | 0.07 (± 0.01) | 0.35 (± 0.01) |
| MULTI-TASK | 95.3 | - | - | 68.3 | - | - |

Table 5: *Average accuracy and forgetting of standard models (left) and joint embedding models (right) on Split CUB. The value of 'OoM' is assigned to a metric when the model fails to fit in the memory. The numbers are averaged across 10 runs using a different seed each time. The results from this table are used to generate Fig 2 in Sec. 6.1 of the main paper.*

| Methods | Split CUB | | |
|---|---|---|---|
| | $A_T(\%)$ | $F_T$ | $LCA_{10}$ |
| VAN | 54.3 (± 2.03) / 67.1 (± 4.77) | 0.13 (± 0.02)/ 0.10 (± 0.04) | 0.29 (± 0.009) / 0.52 (± 0.01) |
| EWC | 54 (± 1.08) / 68.4 (± 4.08) | 0.13 (± 0.02) / 0.09 (± 0.03) | 0.29 (± 0.007) / 0.52 (± 0.01) |
| PI | 55.3 (± 2.28) / 66.6 (± 5.18) | 0.12 (± 0.02)/ 0.10 (± 0.04) | 0.29 (± 0.008) / 0.52 (± 0.01) |
| RWALK | 54.4 (± 1.82) / 67.4 (± 3.50) | 0.13 (± 0.01) / 0.10 (± 0.03) | 0.29 (± 0.008) / 0.52 (± 0.01) |
| PROG-NN | OoM / OoM | OoM / OoM | OoM / OoM |
| **A-GEM (Ours)** | **62** (± 3.5) / **71** (± 2.83) | **0.07** (± 0.02) / **0.07** (± 0.01) | **0.30** (± 0.008) / **0.54** (± 0.02) |
| MULTI-TASK | 65.6 / 73.8 | - / - | - / - |

Table 6: *Average accuracy and forgetting of standard models (left) and joint embedding models (right) on Split AWA. The value of 'OoM' is assigned to a metric when the model fails to fit in the memory. The numbers are averaged across 10 runs using a different seed each time. The results from this table are used to generate Fig 2 in Sec. 6.1 of the main paper.*

| Methods | Split AWA | | |
|---|---|---|---|
| | $A_T(\%)$ | $F_T$ | $LCA_{10}$ |
| VAN | 30.3 (± 2.84) / 42.8 (± 2.86) | **0.04** (± 0.01) / 0.07 (± 0.02) | 0.21 (± 0.008) / 0.37 (± 0.02) |
| EWC | 33.9 (± 2.87) / 43.3 (± 3.71) | 0.08 (± 0.02) / 0.07 (± 0.03) | 0.26 (± 0.01) / 0.37 (± 0.02) |
| PI | 33.9 (± 3.25) / 43.4 (± 3.49) | 0.08 (± 0.02) / 0.06 (± 0.02) | 0.26 (± 0.01) / 0.37 (± 0.02) |
| RWALK | 33.9 (± 2.91) / 42.9 (± 3.10) | 0.08 (± 0.02) / 0.07 (± 0.02) | 0.26 (± 0.01) / 0.37 (± 0.02) |
| PROG-NN | OoM / OoM | OoM / OoM | OoM / OoM |
| **A-GEM (Ours)** | **44** (± 4.10) / **50** (± 3.25) | 0.05 (± 0.02) / **0.03** (± 0.02) | **0.29** (± 0.01) / **0.39** (± 0.02) |
| MULTI-TASK | 64.8 / 66.8 | - / - | - / - |

Table 7: *Computational cost and memory complexity of different LLL approaches. The timing refers to training time on a GPU device. Memory cost is provided in terms of the total number of parameters P, the size of the minibatch B, the total size of the network hidden state H (assuming all methods use the same architecture), the size of the episodic memory M per task. The results from this table are used to generate Fig. 1 and 2 in Sec. 6.1 of the main paper.*

| Methods | Training Time [s] | | | | Memory | |
|---|---|---|---|---|---|---|
| | MNIST | CIFAR | CUB | AWA | Training | Testing |
| VAN | 186 | 105 | 54 | 4123 | P + B*H | P + B*H |
| EWC | 403 | 250 | 72 | 4136 | 4*P + B*H | P + B*H |
| PROGRESSIVE NETS | 510 | 409 | $\infty$ | $\infty$ | 2*P*T + B*H*T | 2*P*T + B*H*T |
| GEM | 3442 | 5238 | - | - | P*T + (B+M)*H | P + B*H |
| **A-GEM (Ours)** | 477 | 449 | 420 | 5221 | 2*P + (B+M)*H | P + B*H |

## F  ANALYSIS OF EWC

In this section we provide empirical evidence to the conjecture that regularization-based approaches like EWC need over-parameterized architectures and multiple passes over the samples of each task in order to perform well. The intuition as to why models need to be over-parameterized is because it is easier to avoid cross-task interference when the model has additional capacity. In the single-pass setting and when each task does not have very many training samples, regularization-based appraches also suffer because regularization parameters cannot be estimated well from a model that has not fully converged. Moreover, for tasks that do not have much data, rgularization-based approaches do not enable any kind of positive backward transfer (Lopez-Paz & Ranzato, 2017) which further hurts performance as the predictor cannot leverage knowledge acquired later to improve its prediction on past tasks. Finally, regularization-based approaches perform much better in the multi-epoch setting simply because in this setting the baseline un-regularized model performs much worse, as it overfits much more to the data of the current task, every time unlearning what it learned before.

We consider Permuted MNIST and Split CIFAR datasets as described in Sec. 6 of the main paper. For MNIST, the two architecture variants that we experiment with are; 1) two-layer fully-connected network with 256 units in each layer (denoted by $-S$ suffix), and 2) two-layer fully-connected network with 2000 units in each layer (denoted by $-B$ suffix).

For CIFAR, the two architecture variants are; 1) ResNet-18 with 3 times less feature maps in all the layers (denoted by $-S$ suffix), and 2) Standard ResNet-18 (denoted by $-B$ token).

We run the experiments on VAN and EWC with increasing the number of epochs from 1 to 10 for Permuted MNIST and from 1 to 30 for CIFAR. For instance, when epoch is set to 10, it means that the training samples of task $t$ are presented 10 times before showing examples from task $t + 1$. In Fig. 7 and 8 we plot the Average Accuracy (Eq. 1) and Forgetting (Eq. 2) on Permuted MNIST and Split CIFAR, respectively.

We observe that the average accuracy significantly improves with the number of epochs only when EWC is applied to the big network. In particular, in the single epoch setting, EWC peforms similarly to the baseline VAN on Split CIFAR which has fewer number of training examples per task.

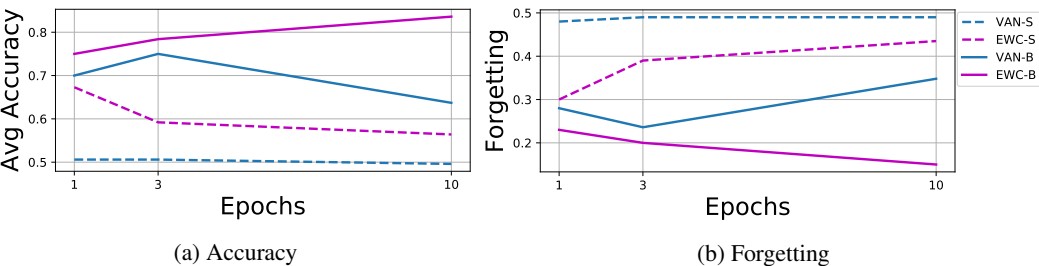

(a) Accuracy           (b) Forgetting

Figure 7: **Permuted MNIST**: *Change in average accuracy and forgetting as the number of epochs are increased. Tokens '-S' and '-B' denote smaller and bigger networks, respectively.*

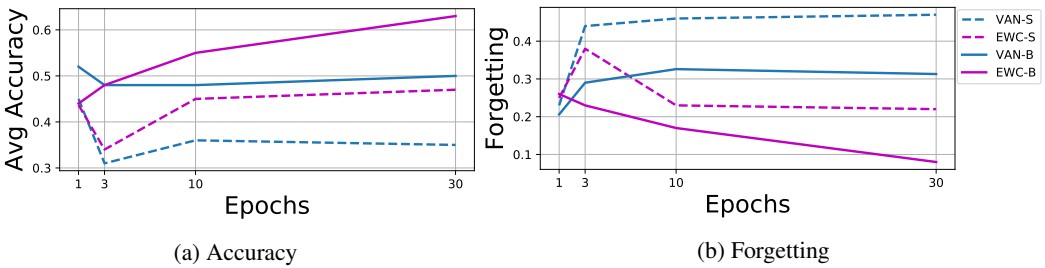

(a) Accuracy           (b) Forgetting

Figure 8: **Split CIFAR**: *Change in average accuracy and forgetting as the number of epochs are increased. Tokens '-S' and '-B' denote smaller and bigger networks, respectively.*

## G  HYPER-PARAMETER SELECTION

Below we report the hyper-parameters grid considered for different experiments. Note, as described in the Sec. 6 of the main paper, to satisfy the requirement that a learner does not see the data of a task more than once, first $T^{CV}$ tasks are used to cross-validate the hyper-parameters. In all the datasets, the value of $T^{CV}$ is set to '3'. The best setting for each experiment is reported in the parenthesis.

- MULTI-TASK
    - learning rate: [0.3, 0.1, 0.03 (MNIST perm, Split CIFAR, Split CUB, Split AWA), 0.01, 0.003, 0.001, 0.0003, 0.0001]
- MULTI-TASK-JE
    - learning rate: [0.3, 0.1, 0.03 (Split CUB, Split AWA), 0.01, 0.003, 0.001, 0.0003, 0.0001]
- VAN
    - learning rate: [0.3, 0.1, 0.03 (MNIST perm, Split CUB), 0.01 (Split CIFAR), 0.003, 0.001 (Split AWA), 0.0003, 0.0001]
- VAN-JE
    - learning rate: [0.3, 0.1, 0.03 (Split CUB), 0.01, 0.003 (Split AWA), 0.001, 0.0003, 0.0001]
- PROG-NN
    - learning rate: [0.3, 0.1 (MNIST perm, ), 0.03 (Split CIFAR, Split AWA), 0.01 (Split CUB), 0.003, 0.001, 0.0003, 0.0001]
- EWC
    - learning rate: [0.3, 0.1, 0.03 (MNIST perm, Split CIFAR, Split CUB), 0.01, 0.003 (Split AWA), 0.001, 0.0003, 0.0001]

- – regularization: [1 (Split CUB), 10 (MNIST perm, Split CIFAR), 100 (Split AWA), 1000, 10000]
- EWC-JE
  - – learning rate: [0.3, 0.1, 0.03 (Split CUB), 0.01, 0.003 (Split AWA), 0.001, 0.0003, 0.0001]
  - – regularization: [1, 10 (Split CUB), 100 (Split AWA), 1000, 10000]
- PI
  - – learning rate: [0.3, 0.1 (MNIST perm), 0.03 (Split CUB), 0.01 (Split CIFAR), 0.003 (Split AWA), 0.001, 0.0003, 0.0001]
  - – regularization: [0.001, 0.01, 0.1 (MNIST perm, Split CIFAR, Split CUB), 1 (Split AWA), 10]
- PI-JE
  - – learning rate: [0.3, 0.1, 0.03 (Split CUB), 0.01, 0.003 (Split AWA), 0.001, 0.0003, 0.0001]
  - – regularization: [0.001, 0.01, 0.1 (Split CUB), 1, 10 (Split AWA)]
- MAS
  - – learning rate: [0.3, 0.1 (MNIST perm), 0.03 (Split CIFAR, Split CUB), 0.01, 0.003 (Split AWA), 0.001, 0.0003, 0.0001]
  - – regularization: [0.01, 0.1 (MNIST perm, Split CIFAR, Split CUB), 1 (Split AWA), 10]
- MAS-JE
  - – learning rate: [0.3, 0.1, 0.03 (Split CUB), 0.01, 0.003, 0.001 (Split AWA), 0.0003, 0.0001]
  - – regularization: [0.01, 0.1 (Split CUB, Split AWA), 1, 10]
- RWALK
  - – learning rate: [0.3, 0.1 (MNIST perm), 0.03 (Split CIFAR, Split CUB), 0.01, 0.003 (Split AWA), 0.001, 0.0003, 0.0001]
  - – regularization: [0.1, 1 (MNIST perm, Split CIFAR, Split CUB), 10 (Split AWA), 100, 1000]
- RWALK-JE
  - – learning rate: [0.3, 0.1, 0.03 (SPLIT CUB), 0.01, 0.003 (Split AWA), 0.001, 0.0003, 0.0001]
  - – regularization: [0.1, 1 (Split CUB), 10 (Split AWA), 100, 1000]
- A-GEM
  - – learning rate: [0.3, 0.1 (MNIST perm), 0.03 (Split CIFAR, Split CUB), 0.01 (Split AWA), 0.003, 0.001, 0.0003, 0.0001]
- A-GEM-JE
  - – learning rate: [0.3, 0.1, 0.03 (SPLIT CUB), 0.01, 0.003 (Split AWA), 0.001, 0.0003, 0.0001]

## H  PICTORIAL DESCRIPTION OF JOINT EMBEDDING MODEL

In Fig. 9 we provide a pictorial description of the joint embedding model discussed in the Sec. 5 of the main paper.

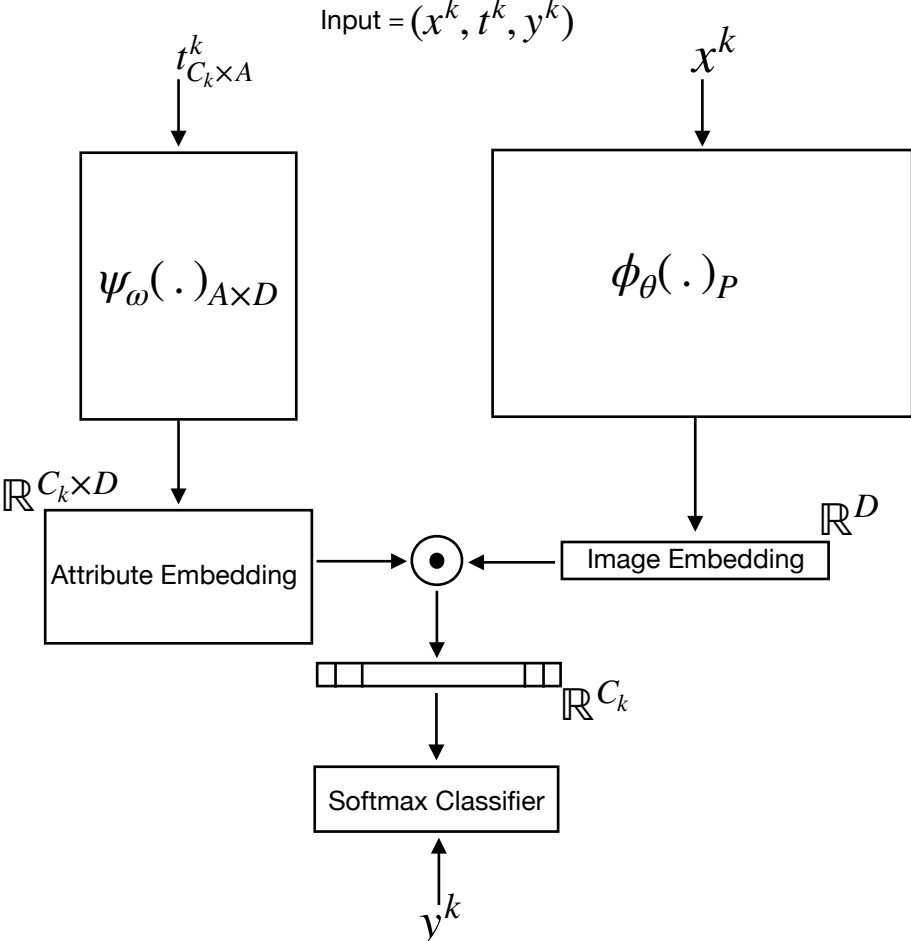

Figure 9: Pictorial description of the joint embedding model discussed in the Sec. 5 of the main paper. Modules; $\phi_\theta(.)$ and $\psi_\omega(.)$ are implemented as feed-forward neural networks with $P$ and $A \times D$ parameters, respectively. The descriptor of task $k$ ($t^k$) is a matrix of dimensions $C_k \times A$, shared among all the examples of the task, constructed by concatenating the $A$-dimensional class attribute vectors of $C_k$ classes in the task.

