# OpenReview forum: "Efficient Lifelong Learning with A-GEM"
_ICLR.cc/2019/Conference_

### Official Review · AnonReviewer2 · 2018-11-02
**There are some interesting ideas here. A more efficient version of GEM.**

**Rating:** 7
**Confidence:** 4

**Review:**

This paper proposes a variant of GEM called A-GEM that substantially improves the computational characteristics of GEM while achieving quite similar performance. To me the most interesting insight of this work is the proof that an inner product between gradients can suffice instead of needing to solve the quadratic program in GEM – which I have found to be a major limitation of the original algorithm.  The additional experiments using task descriptors to enable zero shot learning are also interesting.  Moreover, the discussion of the new evaluation protocol and metrics make sense with further clarification from the authors. Overall, I agree with the other reviewers that this paper makes a clear and practical contribution worthy of acceptance.

---

> ### Author Response · Authors · 2018-11-15
> **The use of task descriptors to expedite learning, LCA and a new train/ eval protocol are important contributions for lifelong learning**
>
> We thank the reviewer for providing the feedback on the draft. Following is our response to the the questions asked  by the reviewer:
>
> Scattered Discourse:
> Our motivation for working on lifelong learning is mostly based on the unprecedented opportunity to learn more quickly new tasks given the experience accumulated in the past. A major reason why catastrophic forgetting is bad is that it prevents the learner from quickly adapting to new tasks that are similar to old tasks.
> The focus of this work is then on sample and computational efficiency in LLL. It is important to impose the restrictions of learning from few examples in a single pass (and to cross-validate on a different set of tasks to properly assess generalization in this single pass setting) as we really aim at models that learn quickly without iterating multiple times over the same data. Moreover, it is important to be able to measure how quickly one learns, and to improve efficiency of existing algorithms (A-GEM and compositional task descriptors).  The current evaluation framework that other works borrow from supervised learning (multiple passes over the data and cross-validate on the same tasks as used for testing) is often misleading, as the methodology (training protocol and metrics) is inadequate for evaluating continual learning algorithms. With this work, we hope to convince the research community to adopt our proposed training/evaluation protocol and to also consider sample/computational and memory efficiency in their metrics.
> We hope the reviewer can find the revised paper more coherent and clear in this respect.
>
> 1, 2: The reviewer is correct in saying that the use of compositional task descriptors and joint embedding models are not specific to A-GEM. In fact, we apply the joint embedding model also to the baseline methods and show improvements on those as well (see fig. 2 and fig. 4). The reason why we introduce them in this work is because a) there may be applications where an agent is given some sort of *description* of the task to perform, and b) since we focus on efficient learning (meaning, learning quickly from few examples), compositional task descriptors enable the learner to perform well at 0-/few-shot learning (see new fig. 5).
>
> 3:
> a) Training/ Evaluation Protocol: To the best of our knowledge, standard practice in LLL is to perform several passes over the data of each task, and several passes over the whole stream of tasks to set hyper-parameters, and then report error on the test set. This evaluation protocol is not adequate because the point of LLL is to quickly learn new tasks, and doing multiple passes over the data defeats the original purpose. Moreover, the prevalent protocol greatly puts the baseline,  which simply finetunes parameters from the previous task without any regularization, at disadvantage. The more the passes are done over the data of a given task, the more the model will forget. Therefore, the conclusions drawn from using the “supervised learning” protocol in a LLL setting can be highly misleading, while using the proposed methodology takes us closer to our goal to fairly assess algorithms in the continual learning setting.
>
> b) LCA: In the few shot learning literature, people specify the number of examples they will be given at test time, and use  $Z_b$ as defined in eq. 4, which is the average accuracy after seeing $b$ minibatches (or a certain number of examples). LCA is the area under the $Z_b$ curve. LCA is a better metric because it also contains information of the values of $Z_j$ for $j <= b$. If $b$ is relatively large, all methods produce similar average accuracy. LCA enables us to distinguish those models that have learned fast, because their 0 or few shot accuracy is higher. Since we care about how quickly a model learns, LCA is a useful metric to assess sample efficiency.
>
> c) Measuring performance on few examples: If by measuring performance on few examples, the reviewer mean reporting $Z_b$ numbers (Eq.4) and not taking the area under the $Z_b$ curve, then we would like to highlight that area under the curve (LCA) is capturing the learner's performance up to the $b$-th minibatch, giving the average profile of the complete few-shot region. $Z_b$, on the other hand, would only give the performance at the current mini-batch and will not have the path information.
>
> 4: Adding error bars: As suggested, we have added the uncertainty estimates measured across multiple runs and seeds in the updated draft. Please take a look at Figs 1, 2 and Tabs. 4, 5, 6. Our conclusions are confirmed.
> Regarding running GEM with task descriptors, we have shown that GEM and A-GEM have similar performance on MNIST and CIFAR. We did not run it on CUB and AWA because GEM is too computationally expensive to run on larger models.

---

> > ### Author Response · Authors · 2018-11-30
> > **Follow-up**
> >
> > May we ask the reviewer, if we were able to address the main concerns that the reviewer had through our rebuttal and revision of the paper? Are there any further issues that the reviewer wants us to address? If so, we would appreciate your feedback and further discussion.

---

> > > ### Comment · AnonReviewer2 · 2018-12-01
> > > **Updated Review**
> > >
> > > Thank you for your detailed rebuttal and revisions to the paper. I do agree that you have addressed my primary concerns and clarified some areas of confusion for me about the paper. I have updated my score in favor of acceptance after the revisions.

---

### Official Review · AnonReviewer1 · 2018-11-02
**Computational improvement of the GEM algorithm for lifelong learning**

**Rating:** 6
**Confidence:** 4

**Review:**

Summary of the paper:

This paper focuses on the problem of lifelong learning for multi-task
neural networks. The goal is to learn in a computationally and memory
efficient manner new tasks as they are encountered while at the same
time remembering how to solve previously seen tasks with a focus on
having only one training pass through all the training data. The paper
builds on the GEM method introduced in the paper "Gradient episodic
memory for continuum learning", NIPS 2017.

The main novelty over the original GEM paper is that A-GEM simplifies
the constraints on what constitutes a feasible update step during its
SGD training so that GEM's QP problem is replaced by a couple of
inner-products (and thus makes A-GEM much more computationally
efficient). This simplification also means that only one gradient
vector (the average gradient computed from the individual gradients of
the task loss of the previously seen tasks) has to be stored at each
update as opposed to GEM where each task specific gradient vector has
to be stored. Thus the memory requirements of A-GEM is much less than
GEM and is independent of the number of already learnt tasks.

The paper then presents experimental evidence that A-GEM does run much
faster and uses less memory and results in performance similar to the
original GEM strategy. The latter point is important as the simplified
A-GEM algorithm - which adjusts the network's parameter to improve
performance on the current task while ensuring the average performance
on the previously seen tasks should not decrease - does not guarantee
as stringently as GEM that the network does not forget how to perform
all the previous tasks.

The paper also introduces an extra performance metric is introduced
  called the "Learning Curve Area" which measures how quickly a new
  task is learnt when it is presented with new material.


Pros and Cons of the paper:

+/- The paper presents a simple intuitive extension to the original GEM
paper that is much more computationally efficient and is thus more
suited and feasible for real lifelong learning applications. And it
shows that performance exceeds other methods that have similar
computational demands. The paper can be viewed as somewhat incremental
but the increment is probably crucial for any real-world practical
application.

+ The validity of the approach is demonstrated experimentally on
  standard datasets in the field.


- Some of the presentation of the material is somewhat vague, in
  particular section 5. In this section a joint embedding model is
  described that helps facilitate zero-shot learning. However, not
  enough detail is given to fully understand or appreciate this
  contribution, see below for details.


Rationale for my evaluation:

The method is somewhat incremental, however, this increment could be
quite practically important. The presentation is lacking in some regard and would benefit
 from some re-working i.e. section 5.


Unclear in the paper:

Section 5 describing the "Joint Embedding Model Using Compositional Task descriptors" is very sparse on detail.  Here are some of the details that I feel are missing:
- In the experiments how is the matrix description (via attributes) of the different tasks $t^k$ learnt/discovered?
- The size of this attribute matrix is able to vary from one task to the next. How does the function $\psi_{\omega}$ deal with this problem?
- What functions are used in the experiments to represent $\psi_{\omega}$?
- In the second last line of paragraph 2 should $C$ be $C_k$? If it should be $C$ how is $C$ chosen?
- In equation (12) should the $c$th column of $\psi_{\omega}$ be extracted as opposed to the $k$th column?

Representative labelled samples from each task are stored in memory
and these are used when learning for a new task. The system
has a fixed memory so when a new task is added then the number of
images stored for each task has to be reduced. Then uniform sampling is
used to randomly decide which images to keep. Could this selection
process be improved upon and would any such improvement have any large
impact on performance?

Typos and minor errors spotted:

In the third paragraph of section 2 it is stated $T^{CV} \ll T$ in the
experiments performed this is not case. I don't think 3 is much less
than 10 or 20.

In figures 4 and 5 it is not entirely clear which curves correspond to
A-GEM and A-GEM-JE from the legend. In the legend the dashed line with
the triangle looks the same the non-dashed line with the triangle. I
presuming A-GEM is the non-dashed line, but only because that makes
things consistent with the previous figures.

---

> ### Author Response · Authors · 2018-11-15
> **Section 5 is clarified. Train/ Eval protocol, new measure for efficiency and the use of task descriptors to expedite learning are novel contributions**
>
> We thank the reviewer for providing the feedback on the draft. Here is our response to the questions asked by the reviewer:
>
> Clarity About Section 5: We have updated the Section 5 of the paper and tried to add additional details about the model. Here are some clarifications:
>
> 1 - Matrix Description t^k: The matrix description is not learnt. It is composed from class attributes.
> For instance, in CUB each class is described by 312 attributes. If the current task has 10 classes, then the task descriptor is a matrix of size 10x312. The task descriptor is the same for all samples belonging to that task. As noted in the section, each input example consists of (x^k, y^k, t^k).
>
> 2 - Variable size of the attribute matrix: Let A be the number of attribute per class (the same across all classes) and C^k the number of classes in task k, then the input task descriptor has size C^k \times A. Module \psi_{\omega} is simply a matrix of size A \times D embedding each attribute. By multiplying the input task descriptor with this embedding matrix, we obtain a matrix of size C^k \times D, embedding each class descriptor. The joint embedding model scores each class by computing a dot product between the image features and the class embeddings (each row of the above matrix), and it turns this scores into probability values using a softmax, as shown in eq. 13.
>
> If the model extracts good image features that reveal the underlying attributes, it can now perform 0 shot learning on unseen classes (as long as their constituent attributes have already been learned for other tasks albeit in different combinations).
>
> In the rewrite of Section 5, we have clarified this point and the corresponding notation.
>
> 3 -  Functions used to represent \psi_{\omega}: We use a lookup table whose parameter matrix has size A \times D.
>
> 4 - Confusion between C and C_k: The reviewer is correct. It should be C_k. We have corrected this in the updated draft.
> 5 - Eq. 12: The reviewer is correct. We have corrected the equation in the updated draft.
>
> Effect of Representative Sampling on the Performance:  [1] showed that using the existing LLL setups and benchmarks, more sophisticated strategies to populate the memory  do not have an appreciable impact. We did try herding-based sampling [2] and got an improved performance of 1-2%. We leave further exploration to future work.
>
> T^{CV} \ll T: In the updated draft, the AWA-10 experiments has been replaced with AWA-20. So, now we have 20 tasks for all the datasets. While, comparatively, 3 may not be much less than 20, in general, the idea is to use a small and separate subset of tasks for the cross-validation which will not be used for further training and evaluation. This allows us to conform to our stricter definition of LLL setting.
>
> Legends of Figs 4 and 5: We have fixed the legend in the updated draft. A-GEM is the one with the dashed line.
>
> [1] RWalk: Riemannian walk for incremental learning: Understanding forgetting and intransigence, ECCV2018.
> [2] Incremental Classifier and Representation Learner: CVPR 2016
>
> *Additional comment*:
> While A-GEM is an important contribution of this paper as it makes the original GEM algorithm much more practical, we believe that the introduction of the new evaluation protocol, new metric and extension using compositional task descriptors are also significant contributions.
>
> Lifelong learning setting entails learning more quickly given the experience accumulated in the past. One reason why catastrophic forgetting is bad is that it prevents the learner from quickly adapting to new tasks that are similar to old tasks.
>
> Since the focus should be on sample and computational efficiency, in this work we considered learning from few examples in a single pass, and cross-validating on a different set of tasks to satisfy that requirement. The metric, the additional efficiency achieved by the use of task descriptors and the new A-GEM algorithm are then part of the same effort to make lifelong learning methods and evaluation protocol more realistic. The current evaluation framework that other works borrow from supervised learning (multiple passes over the data and cross-validate on the same tasks as used for testing) is often misleading. We hope to convince the research community to adopt our proposed training/evaluation protocol and to also consider sample/computational and memory efficiency in their metrics.

---

> > ### Author Response · Authors · 2018-11-30
> > **Follow-up**
> >
> > May we ask the reviewer, if we were able to address the main concerns that the reviewer had through our rebuttal and revision of the paper? Are there any further issues that the reviewer wants us to address? If so, we would appreciate your feedback and further discussion.

---

### Official Review · AnonReviewer3 · 2018-11-04
**A-GEM is a a clear improvement over the previous approach (GEM)**

**Rating:** 7
**Confidence:** 4

**Review:**

The paper is well-written, with the main points supported by experiments.  The modifications to GEM are a clear computational improvement.

One complaint: the "A" in A-GEM could stand for "averaging" (over all task losses) or "approximating" (the loss gradient with a sample).  Both ideas are good.  However, the paper does not address the question: how well does GEM do when it uses a stochastic approximation to each task loss?  (Note I'm not talking about S-GEM, which randomly samples a task constraint; rather, approximate each task's constraint by sampling that task's examples).

Another complaint: reported experimental results lack any associated idea of uncertainty, confidence interval, empirical variation, etc.  Therefore it is unclear whether observed differences are meaningful.

---

> ### Author Response · Authors · 2018-11-14
> **Stochastic version of GEM has very similar run time and memory complexity as the original GEM algorithm**
>
> We thank the reviewer for providing the feedback on the draft. Here is our response to the questions asked by the reviewer:
>
> Q1: We tried the version of GEM where each task loss is approximated by the few examples in the memory for that task as suggested by the reviewer. This approximation yielded slightly better numbers than original GEM:
>
> Method            | DataSet   |   Average Acc   |   Forgetting
> --------------------------------------------------------------------------------------
> Approx-GEM   | MNIST     |   90.1 (+-0.6)   |   0.06 (+-0.01)
>                           | CIFAR      |   61.8 (+-0.5)   |   0.06 (+- 0.01)
> --------------------------------------------------------------------------------------
> GEM                  | MNIST   |   89.5 (+- 0.5)  |   0.06 (+- 0.004)
>                           | CIFAR     |   61.2 (+-0.8)   |   0.06 (+- 0.01)
> --------------------------------------------------------------------------------------
> A-GEM             | MNIST    |   89.1 (+-0.14) |  0.06 (+-0.001)
> (this paper)    | CIFAR      |   62.9  (+-2.2)  |  0.07 (+- 0.02)
>
> However, note that this approximation only makes gradient computation more efficient (although not as much on modern GPUs), but the crux of the computation which is due to the inner optimization problem has the same memory and time complexity as the original GEM; overall, this stochastic version of GEM has very similar run time as the original GEM algorithm. Instead, the proposed A-GEM has much lower time (about 100 times faster) and memory cost (about 10 times lower) while achieving similar performance, as highlighted in the Section 6.1 of the paper.
>
> Q2: As suggested by the reviewer, we have added the uncertainty estimates in the updated draft. As you can see from the updated Figs 1, 2 and Tabs. 4, 5, and 6, conclusions do not change.
>
> *Additional comment*:
> While A-GEM is an important contribution of this paper as it makes the original GEM algorithm much more practical, we believe that the introduction of the new evaluation protocol, new metric and extension using compositional task descriptors are also significant contributions.
>
> Lifelong learning setting entails learning more quickly given the experience accumulated in the past. One reason why catastrophic forgetting is bad is that it prevents the learner from quickly adapting to new tasks that are similar to old tasks.
>
> Since the focus should be on sample and computational efficiency, in this work we considered learning from few examples in a single pass, and cross-validating on a different set of tasks to satisfy that requirement. The metric, the additional efficiency achieved by the use of task descriptors and the new A-GEM algorithm are then part of the same effort to make lifelong learning methods and evaluation protocol more realistic. The current evaluation framework that other works borrow from supervised learning (multiple passes over the data and cross-validate on the same tasks as used for testing) is often misleading. We hope to convince the research community to adopt our proposed training/evaluation protocol and to also consider sample/computational and memory efficiency in their metrics.

---

> > ### Comment · AnonReviewer3 · 2018-11-25
> > **A model of a good author response**
> >
> > I find your response very satisfying and highly professional.  Consequently I am upgrading my review.

---

### Meta-Review · Area_Chair1 · 2018-11-04
**Useful improvement over GEM and a good evaluation methodology**

**Confidence:** 4
**Recommendation:** Accept (Poster)

**Metareview:**


Pros:
- Great work on getting rid of the need for QP and the corresponding proof of the update rule
- Mostly clear writing
- Good experimental results on relevant datasets
- Introduction of a more reasonable evaluation methodology for continual learning

Cons:
- The model is arguably a little incremental over GEM.  In the end I think all the reviewers agree though that the practical value of a considerably more efficient and easy to implement approach largely outweighs this concern.

I think this is a good contribution in this area and I recommend acceptance.